# Multi-Attribute Subset Selection enables prediction of representative phenotypes across microbial populations
Konrad Herbst [1,2,8], Taiyao Wang [3,8], Elena J. Forchielli[2,4,8], Meghan Thommes[2,5,8], Ioannis Ch. Paschalidis [3,5,6,7] ✉ & Daniel Segrè [1,2,4,5,6] ✉

The interpretation of complex biological datasets requires the identification of representative variables that describe the data without critical information loss. This is particularly important in the analysis of large phenotypic datasets (phenomics). Here we introduce Multi-Attribute Subset Selection (MASS), an algorithm which separates a matrix of phenotypes (e.g., yield across microbial species and environmental conditions) into predictor and response sets of conditions. Using mixed integer linear programming, MASS expresses the response conditions as a linear combination of the predictor conditions, while simultaneously searching for the optimally descriptive set of predictors. We apply the algorithm to three microbial datasets and identify environmental conditions that predict phenotypes under other conditions, providing biologically interpretable axes for strain discrimination. MASS could be used to reduce the number of experiments needed to identify species or to map their metabolic capabilities. The generality of the algorithm allows addressing subset selection problems in areas beyond biology.

As the price of DNA sequencing keeps decreasing, reading genomes is no longer the limiting factor in understanding an organism: interpreting genomes is. In principle, genomes carry information on how an organism will behave in different environments, but in practice, the interpretability of genomic information is severely limited by our lack of knowledge about the function of many individual genes[1,2] and the interactions between genes and environments[3,4], as well as by a limited understanding of how genes work together to produce higher-level functions[5,6]. The challenge of predicting organismal functions from genomes is particularly relevant in microbes, where systems biology approaches provide a potential avenue for building mechanistic models of metabolism at the genome-scale[7,8]. Despite constant improvements in these genome-to-phenotype inferences, there is rising interest in exploring a complementary avenue to characterizing and understanding organismal behavior starting from large-scale phenotypic data. Through this approach, also referred to as phenomics, insight and knowledge are sought from the measurement of phenotypes across many organisms (or genetic perturbations) and environmental conditions[9–13]. Phenomics approaches the challenge of understanding biological systems from a top-down data-driven perspective: phenotypes are measured, often in a high-throughput manner, and then analyzed in search for associations (between genotypes and phenotypes, or between different phenotypes)[14–21]. In turn, these associations can be used to understand how the individual subsystem properties, give rise to cell-, organism- or community-level functions, including mutual interdependencies in microbial communities[22–24]. Furthermore, phenotypic data play an important role in gene annotation[25], and mechanistic model testing and refinement[26].

The development and applicability of phenomics approaches is crucially dependent on the capacity to reduce measurement costs, and to appropriately analyze high-throughput phenotypic data of microorganisms grown in a variety of environments[25,27,28]. A lot of work has been done to characterize the phenotypes of microbes for applications ranging from drug discovery to industrial fermentation[29,30]. Numerous technologies have been developed to facilitate the generation of phenotypic data in microbes and other biological systems[31], including Biolog Phenotype MicroArrays[12,32], robotic screening tools[33], microfluidics devices[34–36], imaging[20,37,38] and multiplexed bioreactors, such as the eVOLVER[39]. However, compared to the

[1]Bioinformatics Program, Boston University, Boston, MA, USA. [2]Biological Design Center, Boston University, Boston, MA, USA. [3]Division of Systems Engineering, Boston University, Boston, MA, USA. [4]Department of Biology, Boston University, Boston, MA, USA. [5]Department of Biomedical Engineering, Boston University, Boston, MA, USA. [6]Faculty of Computing and Data Science, Boston University, Boston, MA, USA. [7]Department of Electrical and Computer Engineering, Boston University, Boston, MA, USA. [8]These authors contributed equally: Konrad Herbst, Taiyao Wang, Elena J. Forchielli, Meghan Thommes. ✉e-mail: yannisp@bu.edu; dsegre@bu.edu

rapid decrease in sequencing cost, measuring phenotypes is still relatively cost and labor-intensive; thus, prioritizing which conditions and organisms to measure is a crucial component of experimental design. Ideally, one would want to narrow down a set of measurements to those organisms and conditions that are most informative for reconstructing larger phenotype matrices and provide the most predictive power regarding other measurable phenotypes. For biologists, focusing on those phenotypes of the highest informational value would have two major advantages: (1) reduce the experimental burden of large screens, and (2) provide insight into mechanistic links between traits.

Numerous statistical approaches have been employed to help extract the most informative subsets of phenotypic datasets[40–42]. Classical methods, such as clustering and latent variable methods (including PCA), are often used as dimensionality reduction tools for data interpretation and visualization purposes[43]. However, those methods do not provide a clear identification of a subset of variables such that the whole dataset can be quantitatively expressed as a function of those selected variables. Conversely, regression analysis can be used to build predictive models based on a specific set of attributes[44]. However, regression typically relies on prior knowledge of which attributes should serve as predictors and which model is used to represent the relation between predictor and response attributes, limiting their utility for phenotype organization, where no a priori distinction between predictor and response variables exists.

We approach the identification of the most informative phenotypes by asking the following question: Which subset of phenotypes of a given subset size describes the remaining phenotypes best? We formulate a mathematical framework to answer this question, in which phenotypes are classified into sets of predictor and response attributes. At the same time, generalized linear regressions of those sets are used to identify the most descriptive phenotype set. Exhaustive enumeration of all possible sets of predictor and response attributes gives rise to an extremely large number of possible solutions. Nevertheless, efficient training of the regression models can be achieved by using mixed integer linear programming (MILP). This algorithm, to which we refer as "multi-attribute subset selection" (MASS), finds solutions explicitly representing the most informative phenotype sets, which to our knowledge, constitutes a new approach to the phenotype selection problem.

Over the last decade, dramatic improvements in MILP solvers in conjunction with the widespread availability of high-dimensional biological datasets have created attractive opportunities to expand the applications of MILP to questions of biological importance[45]. Great strides have been made in the analysis of DNA sequences[46], protein–protein interaction networks[47], and mass spectrometry data[48] using linear programming methods. However, the preceding examples modeled a single response attribute vector as a linear combination of predictor attributes; methods capable of modeling multiple attributes as the combination of other phenotypes require a novel formulation and have to our knowledge, not been attempted.

MASS is inspired by the application of MILP to solve the subset selection problem[49,50] and exploits several heuristics that can substantially speed up the computational time needed to find an optimal solution, similar to the process outlined in ref. [51]. We applied MASS to three microbial phenotype datasets, and successfully identified sets of predictor attributes with the highest predictive power. MASS has the potential to enable experimentalists to minimize the number of experiments they need to perform while maintaining, with a certain degree of uncertainty, the most information regarding a microbe's phenotype. MASS and its possible extensions could find broad applicability in other data-rich scientific areas beyond biology.

## Results
### A method to separate growth phenotypes into predictors and responses
We designed MASS with the goal of exploring large phenotypic datasets. Specifically, we want to choose the most informative subset of the environmental conditions and aim at using phenotypes under such conditions as predictors of phenotypes under all the remaining conditions. The challenge

of this question is that it involves two steps that are usually performed in separate calculations. In the first step, we would typically choose which phenotypes are predictors (independent variables), and which are the response (dependent variables). In a second step, we would perform a linear regression, i.e., find coefficients necessary to compute the response phenotypes as linear combinations of the predictor phenotypes. What makes our approach mathematically challenging, is that we do not know or assume a priori which phenotypes will be predictors and which ones will be responses. The MASS algorithm makes it possible to pursue both steps concurrently. In other words, the algorithm explores the many possible choices of predictors and simultaneously identifies the ones such that regression done using those predictors gives the best estimate of the responses. This algorithm involves both integer variables, describing which phenotypes are chosen as predictors, and continuous variables, which capture the regression coefficients, and is thus implemented using a mixed integer linear programming (MILP) approach.

The formulation of MASS as a combinatorial optimization problem is described in detail in the Methods and illustrated in Fig. 1. Inputs to MASS include a phenotype matrix ($X$) of $n$ organisms by $m$ environmental conditions. For each possible number of predictors $p$, MASS selects $p$ predictors from $m$ environmental conditions to predict $m - p$ responses using linear regression. The outputs of MASS consist of a binary predictor vector ($z$) whose elements indicate whether an environment is a predictor or response. Note that as we increase the number of predictors $p$ for a given dataset, a condition that is selected as a predictor for a given $p$, may be selected to be in the response set for a different $p$.

In our case, the data consists of discrete growth phenotypes. While MASS provides optimal choices of predictors based on linear regression, we wanted to check whether those same predictors would also be useful as variables for the classification of the data based on an alternative methodology (see Methods). For each dataset, we trained random forest (RF) models[52–54] using the predictors determined by MASS. RF models can have multiple predictors and one response variable. Therefore, the number of RF models trained depends on the number of predictors to which MASS has been constrained. Consider, for example, a dataset containing 11 conditions. If the number of predictors $p$ is set to 1, then there will be $11 - 1 = 10$ responses, and 10 random forest models need to be trained, one for each response. Likewise, when $p$ is set to 10 predictors, there will be $11 - 10 = 1$ response, and only one random forest model has to be trained for the one response. Prior to training the models, we split the data into training and test sets. We then calculated a score for the test sets to estimate the performance of the random forest. We chose the Matthews Correlation Coefficient (MCC) as a performance score able to deal with the potential imbalance in data categories[55] (see Methods). While random forests are used as a practical approach in this study to validate our MASS analysis, we would like to highlight that MASS essentially is a feature selection technique, and we envision that the resulting predictor sets can be agnostically used with any other supervised learning method (Fig. 1).

We apply MASS to three distinct datasets. The first dataset (DATASET 1) is a matrix of growth phenotypes (optical density (OD)) for 65 marine heterotrophic bacteria on 11 different types of carbon sources (see ref. 21 and Fig. 2). The second dataset (DATASET 2), considerably larger (637 bacteria for 46 carbon sources), is a subset of the comprehensive BacDive database of microbial phenotypes[56] including fermentation phenotypes on different carbon sources. The third dataset (DATASET 3) is a compendium of discrete growth phenotypes collected in a practical guide for the identification of different yeast species based on their growth capacity under different conditions[27], and includes more complex phenotypes (three growth phenotypes, for 462 yeast strains over 38 conditions).

### Analysis of DATASET 1: growth of marine heterotrophic bacterial on well-defined media
We first applied MASS to a dataset of limited size (DATASET 1), which resulted from a characterization of marine heterotrophic bacteria grown on the different carbon sources contained in a standard marine broth medium

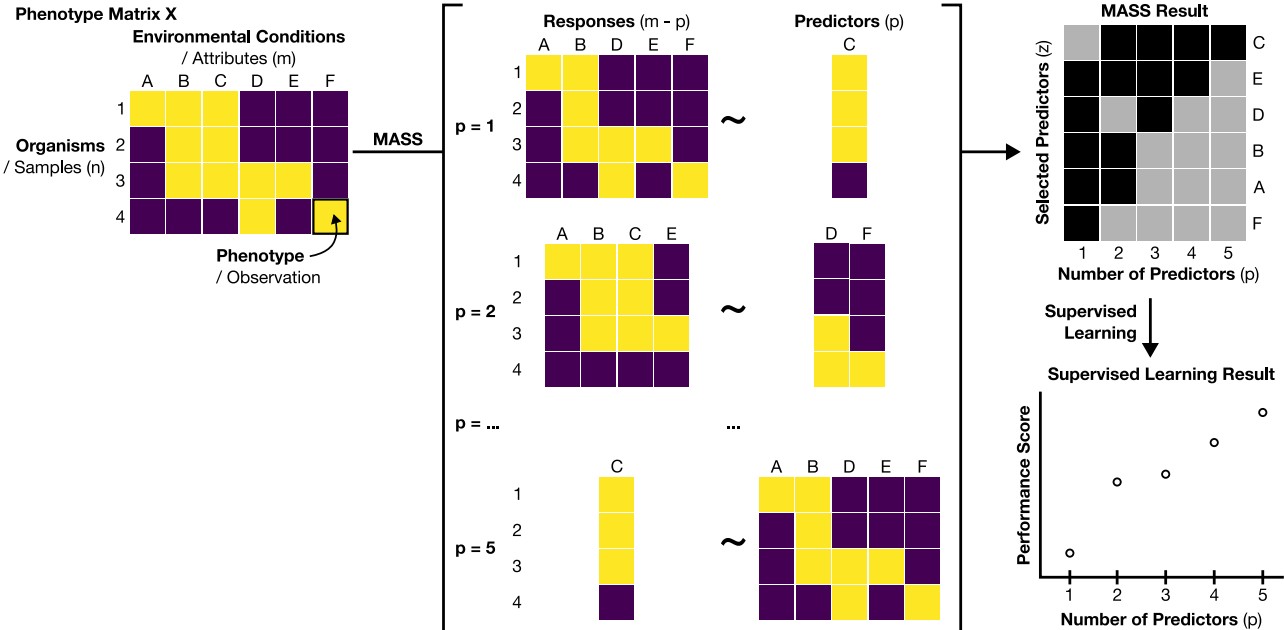

**Fig. 1 | Overview of the multi-attribute subset selection (MASS) approach.** MASS takes as input a matrix (**X**) of observations of samples by attributes, in this case the phenotypes of **n** organisms under **m** different environmental conditions. For each fixed number of predictor variables, **p**, MASS provides as output a binary vector **z** indicating the predictor variables that predict the remaining response variables with the highest accuracy. Subsequently, the labeled data can be used to build models using supervised learning methods, such as random forest models, as done in this study.

(Fig. 2a)[21]. As illustrated in Fig. 2b, MASS provides the predicted optimal choice of predictor media conditions as a function of the number of predictors. In this case, if only one predictor is allowed ($p = 1$), the MASS algorithm selects the "oligosaccharide" condition. For $p = 2$, the chosen representative conditions are "neutral sugars" and "peptides", supplemented by "amino acids" when $p = 3$.

We noted that certain conditions were selected as predictors more often than others (Fig. 2a) and hypothesized that the frequency a condition is selected as a predictor by MASS corresponds with its information content (Fig. 2c). To test this hypothesis, we calculated the Shannon Entropy as a measure for the information content of individual conditions and noted that it correlates with the number of times a condition is selected as a predictor by MASS (Fig. 2b, c and Supplementary Fig. 1a). However, conditions selected most often are not necessarily associated with the highest entropy. This is further confirmed by the random forest prediction based on conditions selected by MASS (Fig. 2d), which shows that MASS performs better than a predictor condition set selected based on maximal entropy. As a control, we also trained random forest models with a random choice of predictor conditions. These models performed considerably worse than the MASS or maximum entropy predictor set selections, highlighting the importance of careful predictor set selections (Fig. 2d and Supplementary Fig. 3). As shown later, these results still hold in principle for larger datasets.

In order to obtain further insight into the implications of the MASS results for the biological systems under study, we also inspected, for each response individually, how well that response phenotype could be predicted by random forest models trained with the predictor sets selected by the MASS algorithm (Supplementary Fig. 2). Interestingly, "amino sugars" are well predicted even when considering only the top three predictors selected by MASS ($p = 3$) The respective predictor conditions contain "neutral sugars", "peptides", and "amino acids". This suggests potential overlaps in metabolic pathways for the utilization of these carbon sources. In contrast, growth under the medium condition containing "organic acid" was difficult to predict, suggesting that this trait is not linked to any particular set of carbon sources tested in the experiment. Altogether, these findings are in line with the previously reported biological interpretation of this and similar datasets, which demonstrated how the different bacteria can be broadly subdivided into major subgroups associated with sugars vs. amino acid/peptide vs. organic acid substrates, respectively[21,57].

## Analysis of DATASET 2: bacterial fermentation on different carbon sources

To evaluate the scalability and performance of MASS on a larger dataset, we identified phenotypic matrices of larger size and complexity, and for which no prior analysis of this kind had been implemented. BacDive is a large database collecting a wide range of phenotypes and metadata for several thousand organisms deposited at the German Collection of Microorganisms and Cell Cultures (DSMZ)[56]. We focused here on a complete subset of this data (see also Methods), which amounts to a matrix of fermentation phenotypes for 637 microbial species on 46 different carbon sources as environmental conditions (Fig. 3a).

The predictors selected by MASS with increasing numbers of predictors $p$ provide a ranking by importance which might be informative about the underlying biological system. We tested this conjecture by dissecting the first couple of conditions which have been selected by MASS as most descriptive (Fig. 3e). We mapped the fermented carbon sources to their respective monomers where applicable (Fig. 3f) and highlighted their entries into glycolysis as part of the central carbon catabolism (Fig. 3g). When one predictor was allowed ($p = 1$), MASS selected cellobiose, a disaccharide consisting of two glucose monomers. Glucose enters glycolysis at the very beginning of the pathway. When two predictors were allowed ($p = 2$), in addition to cellobiose, MASS selected raffinose, a trisaccharide which consists of the monomers galactose, fructose, and glucose. This opened an additional entry point through fructose into glycolysis, while galactose enters glycolysis through glucose-6-phosphate as an alternative to glucose. Raffinose was substituted by lactose and arabinose when three predictors were allowed ($p = 3$), opening an entry point via fructose-1,6-diphosphate as an alternative to fructose. Allowing for four predictors ($p = 4$) apparently resulted in major changes in the selection of the most informative substrates by MASS, but mapping the selected substrates to their respective monomers revealed that just one additional entry point into glycolysis was opened through arbutin. This pattern of incremental addition of entry points into glycolysis continued as more and more predictors were allowed to be

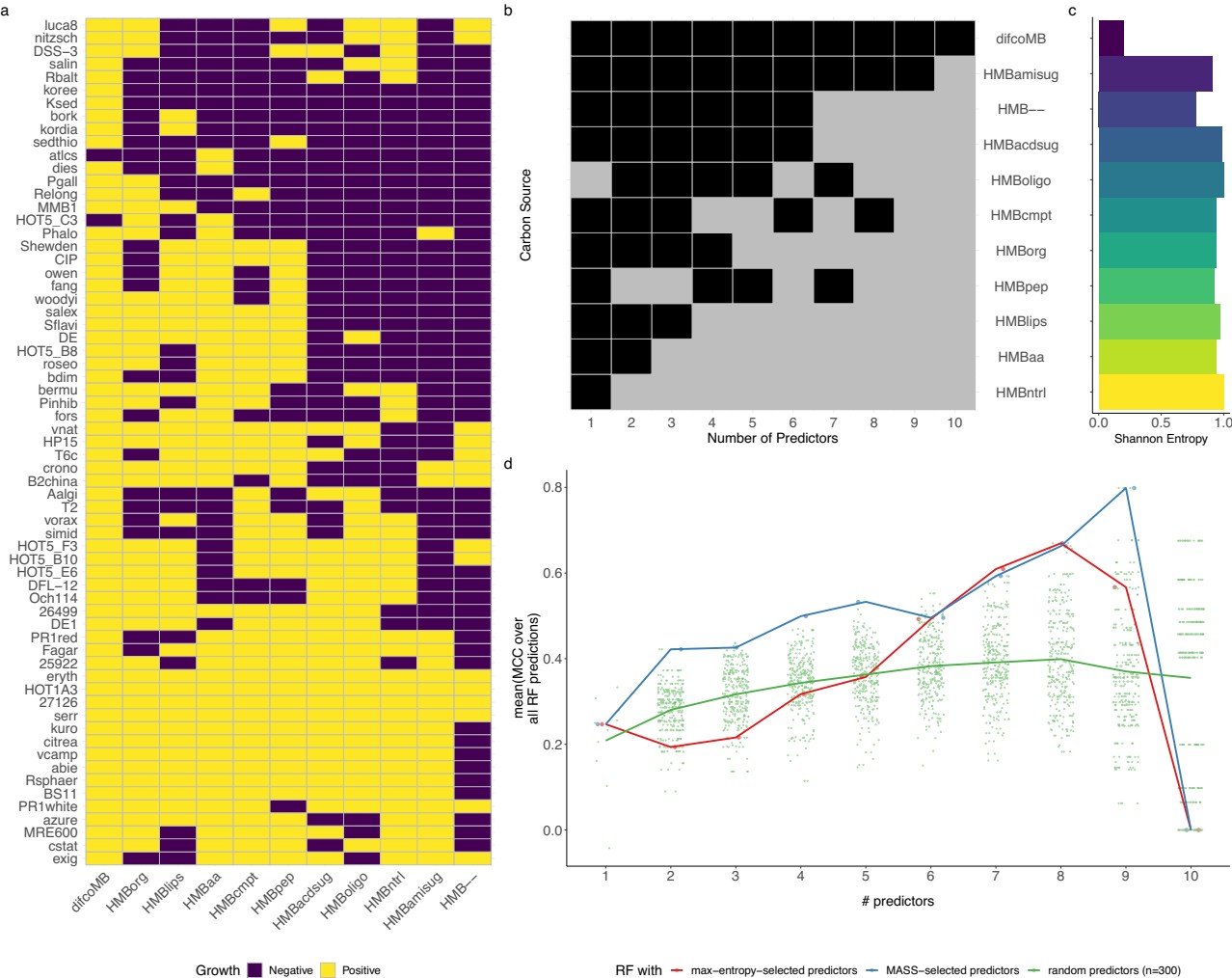

**Fig. 2 | MASS successfully identifies predictor features for a small microbial growth dataset. a** Sixty-five different marine heterotrophic bacterial strains (rows) were grown individually on 11 different media (columns) (see ref. 21 for details) including Difco Marine Broth (difcoMB), eight engineered media with single classes of carbon sources (HMBpep = peptides; HMBaa = amino acids; HMBlips = lipids; HMBoligo = oligosaccharides; HMBorg = organic acids; HMBntrl = neutral sugars; HMBamisug = amino sugars; HMBacdsug = acidic sugars), a defined medium containing all eight carbon classes (HMBcmpt), and a medium with no added carbon sources (HMB−). The names of the different strains are available in Supplementary Data 1. **b** Matrix showing which media were used as a predictor (gray) or as a response (black) as a function of the total number of predictors allowed (parameter $p$). **c** Shannon entropy of each medium. **b**, **c** Media are arranged in descending order of how frequently they were used as predictors. **d** Average Matthews correlation coefficient (MCC) of random forest classifiers for each number of predictors, $p$. The classifiers were trained either using the MASS selection of predictors (blue), predictor sets selected based on maximum Shannon entropy (red), or 300 random draws of conditions used as predictors (green). Each point represents the mean MCC obtained via fivefold cross-validation; the thick lines are the mean of those means across all MCC values for a respective $p$. Source Data for Fig. 2c, d is available in Supplementary Data 4.

selected by MASS (Fig. 3e–g). In summary, we see that with an increasing number of predictors, MASS selected carbon sources covering an increasing number and tiers of entry points into glycolysis (Fig. 3g). This indicates that the predictor selection based on the fermentation phenotype data recapitulates potential topological constraints on central carbon catabolism. These results highlight how the application of MASS on suitable datasets has the potential to reveal more fundamental principles underlying the biological system under study.

Conditions which were selected by MASS more frequently were also more likely to have a higher Shannon Entropy, similarly as observed before for DATASET 1 (Fig. 3b, c and Supplementary Fig. 1b). This suggests that using high-entropy conditions would lead to models which are equally predictive as when trained using predictor conditions resulting from MASS. Indeed, as shown in the analysis of random forest performance (Fig. 3d), entropy-based choice of attributes worked much better than when the predictor conditions were selected randomly. However, the choice suggested by MASS still outperforms an entropy-based selection. This is likely because MASS selects multiple attributes for $p > 1$ that are jointly the most informative ones, while entropy is associated with individual conditions. Thus, MASS may recapitulate the complex structure of the dataset, and account for correlations across phenotypic vectors, going beyond the capabilities of an entropy criterion.

Of note, some conditions have very unbalanced fermentation phenotype frequencies reflecting either rare or ubiquitous phenotypes. Those phenotype vectors were generally selected last by MASS, highlighting how unbalanced phenotype vectors are poor descriptors for other phenotypes. Generally, we observed that the mean performance of the random forest classifiers increased with the number of predictors, irrespective of the performance measure chosen to evaluate the classifiers (Supplementary Fig. 4). This trend ends after 27 or 33 predictors for predictor selections based on Shannon entropy or MASS respectively coinciding with the transition from conditions of high to low informational value. We interpret this as an indication that the available information within a dataset is used by the MASS predictor selections most efficiently (Fig. 3d).

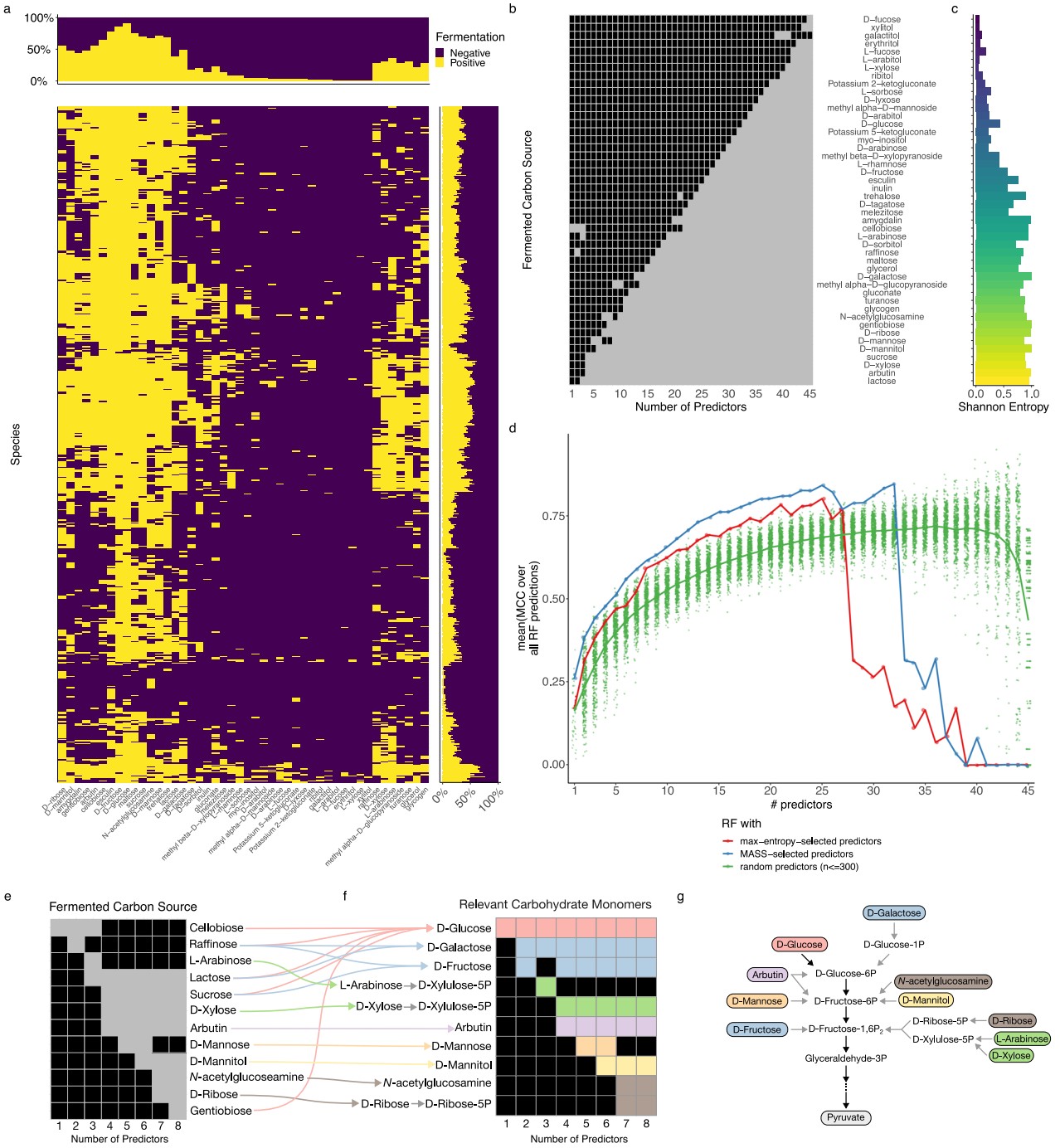

**Fig. 3 | MASS is applicable for larger datasets and recapitulates biological understanding. a** Fermentation phenotypes of 637 bacterial species (rows) grown on 46 different carbon sources (columns) downloaded from BacDive. Fermentation of a specific carbon source is indicated in yellow; and negative (no) fermentation is indicated in purple. Marginal bar charts summarize the phenotype frequency for each species or each carbon source respectively. **b** Matrix showing MASS result in which carbon sources were used as a predictor (gray) or as a response (black) as a function of the total number of predictors allowed (parameter *p*). **c** Shannon entropy of each carbon source. **b**, **c** Media are arranged in descending order of how frequently they were used as predictors. **d** Average Matthews correlation coefficient (MCC) of random forest classifiers for each number of predictors, *p*. The classifiers were

trained either using the MASS selection of predictors (blue), predictor sets selected based on maximum Shannon entropy (red), or 300 random draws of conditions used as predictors (green). Each point represents the mean MCC obtained via fivefold cross-validation; the thick lines are the mean of those means across all MCC values for a respective *p*. **e** Increasing the number of fermented carbon sources selected as predictors with MASS (focusing on up to *p* = 8 predictors) reveals a hierarchy of most descriptive carbohydrate monomers (**f**). **g** Those monomers enter central carbon metabolism via different routes. Only the relevant parts of glycolysis are shown, and reactions are differentiated into direct, single-step (black arrows) and indirect, multi-step reactions (gray arrow). Source Data for Fig. 3c, d is available in Supplementary Data 4.

## Analysis of DATASET 3: yeast carbon assimilation growth profiles

Following the application of MASS to two binary datasets, we were interested in evaluating its performance on a categorical dataset that included more than two possible response values. We used a subset of a vast phenotypic resource describing the aerobic growth of 462 yeast species on 44 different compounds as the sole carbon sources[27]. This reference manual was written mainly as a practical guide to identify yeasts using phenotype profiles. The idea is that, given an unknown yeast species, one would grow it on a predefined series of conditions, gradually narrowing down the options for its taxonomy, and ultimately resulting in a unique identification. Our algorithm has the potential to produce a sequence of maximally informative conditions that would allow one to solve this problem in a general, unsupervised fashion.

Each carbon source and yeast species exhibit different profiles (Fig. 4a). Some carbon sources (e.g., glucose) could be utilized by almost all species, while other carbon sources (e.g., methanol and inulin) could only be utilized by very few species (Fig. 4a). Likewise, some species exhibited "generalist" tendencies, since they were able to grow on most carbon sources, while

others were "specialists", as they could only grow on a small portion of carbon sources (Fig. 4a).

We used MASS to determine which carbon sources best predict yeast growth on other carbon sources. In case only one predictor was allowed ($p = 1$), D-xylose was selected as the most descriptive predictor. This compound is a major component of lignocellulosic material. As $p$ increases (for all $p > 1$), D-gluconate, an acid frequently found in fruit, honey, and wine[58], becomes a prominent and ubiquitous predictor (Fig. 4b). The next two important predictors appearing after D-xylose and gluconate are maltose (a disaccharide degradable into glucose) and glycerol, which feeds into the middle of glycolysis. As observed for DATASET 1 and 2, the larger the entropy of a carbon source, the more often it was used as a predictor (Fig. 4b, c and Supplementary Fig. 1c). However, once again, upon implementing a random forest model based on the MASS choice of predictors at different values of $p$, the predictors chosen by MASS perform better than both random choices and predictors ranked by entropy (Fig. 4d and Supplementary Fig. 5).

We could observe that the performance generally increased with an increasing number of predictors $p$ for up to 16 predictors, and training with

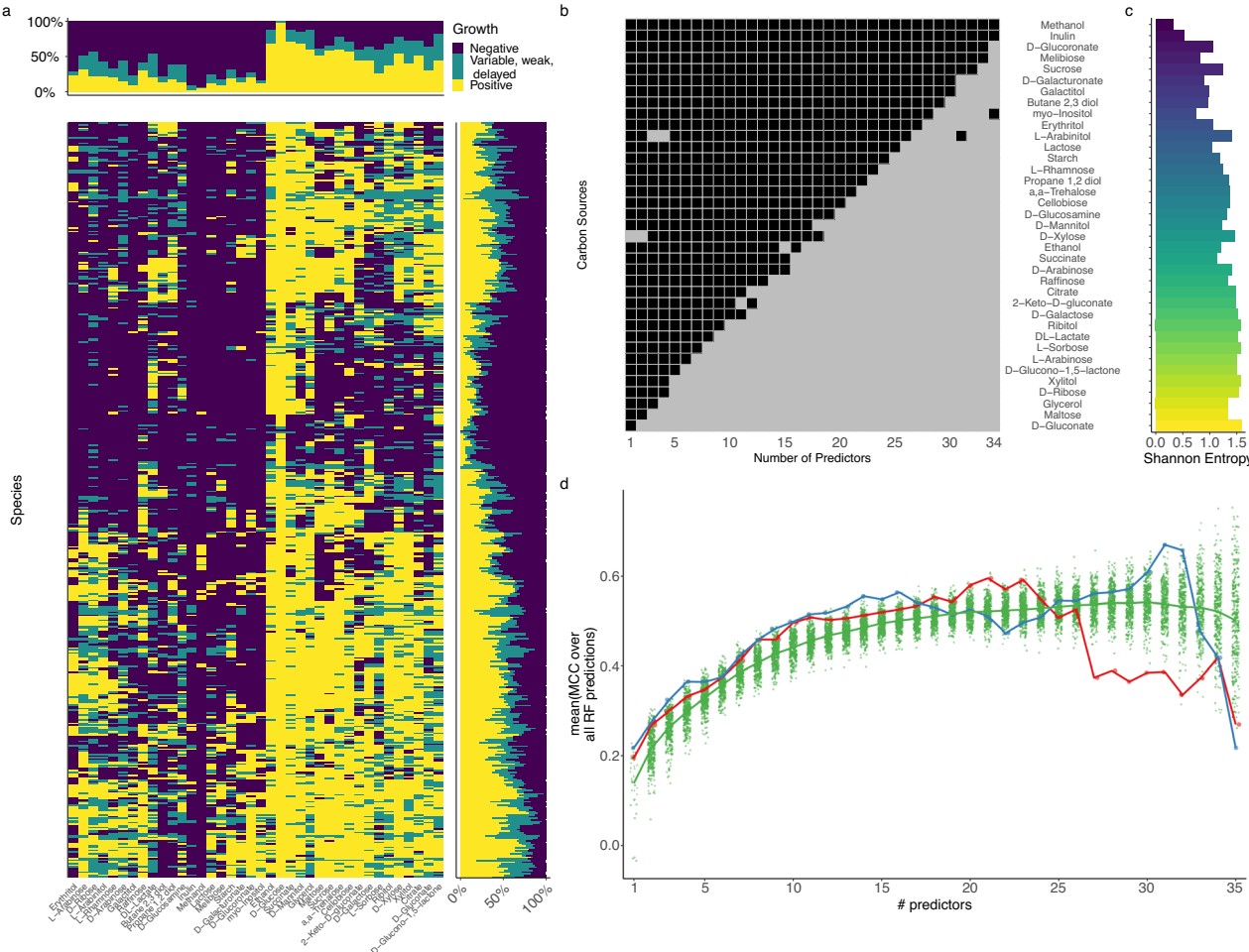

**Fig. 4 | MASS is applicable for categorical data. a** Growth phenotypes of 492 yeast species (rows) grown on 39 different carbon sources (columns)[27]. Positive growth on a specific carbon source is indicated in yellow; variable, weak, or delayed growth is indicated in teal; and negative (no) growth is indicated in purple. Marginal bar charts summarize the phenotype frequency for each species or each carbon source respectively. **b** Matrix showing MASS result in which carbon sources were used as a predictor (gray) or as a response (black) as a function of the total number of predictors allowed (parameter $p$). Growth in glucose was excluded from the MASS analysis. **c** Shannon entropy of each carbon source. **b, c** Media are arranged in

descending order of how frequently they were used as predictors. **d** Average Matthews correlation coefficient (MCC) of random forest classifiers for each number of predictors, $p$. The classifiers were trained either using the MASS selection of predictors (blue), predictor sets selected based on maximum Shannon entropy (red), or 300 random draws of conditions used as predictors (green). Each point represents the mean MCC obtained via threefold cross-validation; the thick lines are the mean of those means across all MCC values for a respective $p$. Source Data for Fig. 4c, d is available in Supplementary Data 4.

predictors selected through MASS resulted in the best-performing classifiers (Fig. 4d). This indicates that MASS succeeds in identifying the most relevant environmental conditions in phenotypic datasets and can handle categorical data beyond two classes.

## Discussion

The problem of attribute or feature selection is not new to biology, and considerable work has been done to improve model performance on high-dimensional biological datasets by separating the most relevant variables from those that may be uninformative, irrelevant, or redundant[42,59,60]. One traditional response to this challenge can be to use dimensionality reduction using methods such as PCA, but these approaches tend to obscure interpretability by creating latent variables, complicating downstream analysis[61]. Feature selection procedures are usually employed in a supervised learning framework for which class labels must be known a priori[40–43]. To the best of our knowledge, MASS is the first method that optimally selects a subset of predictor and a complementary set of response attributes from a dataset using a MILP framework. A key aspect of the MILP approach is that it simultaneously optimizes the regression coefficients which are continuous variables, and the integer variables that identify which features should be used as predictors. In contrast, for example, each principal component in PCA is a linear combination of all available attributes, and it would be less straightforward to choose the most descriptive attributes out of those. As opposed to other methods that broadly fall into the category of learning problems (such as Multi-Modal Best Subset (MOSS) modeling, which is focused on product quality optimization based on sensors in additive manufacturing[62]), our approach is specifically helpful for analyzing biological phenotype matrices as it is designed to find, given a certain number of attributes ($p$), a selection of predictor conditions (attributes) which optimally predicts all the remaining attributes.

It is important to note that while MASS selects the optimal subset of features to explain the remaining features, it does not optimize the predictability of any individual feature. In the future, it may be interesting to explore our formulation when using tree-based models for the regression rather than linear regression[63]. Furthermore, in this work, MASS was applied solely to discretized fitness data; future developments could extend the current method to more general cases, including continuous variables. This is a particularly interesting prospect, as it would add a layer to the biological interpretation of microbial phenotypes by considering, for example, the magnitude of growth in addition to the ability to grow under various conditions.

Our results suggest that MASS has powerful applications for large-scale phenotypic screens by greatly reducing the experimental burden. By identifying combinations of phenotypes that best predict other phenotypes within a dataset, MASS can help prioritize large-scale phenotyping efforts. After performing an initial phenotypic screen for a limited number of samples, MASS would allow one to select the subset of most descriptive conditions to be used for phenotyping a larger number of samples, based on available resources and the desired level of accuracy. Here, we focused on microbial phenotypes in different environmental conditions, but it is not difficult to imagine other contexts in which MASS could be applied: examples include predicting antibiotic sensitivity of different bacterial strains, optimizing metabolite fluxes for biotechnological applications, and mapping inter-microbial interactions in a natural or synthetic microbial community. A particularly exciting potential application of MASS is in the selection of drug candidates for in vivo studies and ex vivo drug screening applications[64,65]. Reducing the enormous experimental burden required to execute these assays would represent an important outcome for many researchers, likely to advance the pace of biological discovery, and to facilitate the commercial feasibility of large-scaled screening assays.

From a biological perspective, the results of MASS could also help to interpret the resulting data. The fact that a given phenotype can be expressed as a linear combination of other phenotypes may suggest an underlying mechanistic relationship between these phenotypes. In the analysis of the marine dataset (DATASET 1), some phenotypes can be well predicted from other phenotypes, suggesting that these metabolic traits are linked for mechanistic or ecological reasons. For example, growth on amino sugars is predicted very well by using growth on neutral sugars, amino acids, and peptides as predictors. This suggests that overlapping metabolic pathways are utilized in the assimilation of these carbon sources. Conversely, growth on the organic acids was difficult to predict through other phenotypes, suggesting that this trait is not linked to any particular set of carbon sources tested in the experiment[21,57]. In another example (DATASET 2), the most descriptive conditions of carbon sources selected by MASS for fermentation seem to reveal a hierarchy of entry points into glycolysis hinting at fundamental constraints acting on the structure of central carbon catabolism.

An interesting finding of our analysis is the fact that, while predictability increases with the number of predictors ($p$) until a certain level ($p < 15$), the performance score drops abruptly as $p$ approaches the total number of variables (see Figs. 3d and 4d). This is because MASS is designed to select all the most descriptive attributes first, leaving for last the least descriptive ones. In other words, for large $p$, the $m$-$p$ response attributes which MASS can still select cannot be easily predicted from the predictor attributes. While exploring very large $p$ values is of little practical utility anyway, we thought it may be interesting to understand whether the specific threshold may relate to some biological properties of the datasets. One can note that both in DATASETS 2 and 3, numerous conditions resulted in positive phenotypes (i.e., observed growth or fermentation) across many of the species. We hypothesize that the substrates used in those conditions might belong to core metabolic pathways evolutionarily conserved across species. In contrast, other conditions rarely resulted in positive phenotypes and therefore might relate to auxiliary metabolic pathways. Interestingly, the number of conditions resulting in ubiquitous phenotypes seems to coincide with the number of allowed attributes ($p$), beyond which the performance starts to drop for random forest models trained with attributes selected by MASS (Figs. 3d and 4d). This suggests that the specific structure of our datasets, and specifically the nature of core vs. auxiliary metabolic pathways may contribute to the observed drop in performance score.

In summary, we present MASS as an algorithm to partition phenotype vectors into a set of most descriptive predictors and respective responses. Phenotypic information processed through MASS will allow researchers to prioritize efforts on high-value phenotypes or shift resource allocation to ensure the most important data is collected. Notably, our generic problem formulation and the MILP framework underlying MASS are not restricted to the particular type of data explored in this study. It would be, therefore, valuable to explore the usability of this approach in other fields of research and practical applications.

## Methods
### Data pre-processing
**Dataset 1**. This dataset was previously acquired in our lab and published elsewhere[21]. The data were downloaded from the data analysis repository of the original study (https://github.com/segrelab/marine_heterotrophs/blob/main/data/growth_profiles.xlsx). The resulting growth phenotype matrix was discretized by assigning the integer 1 to non-zero (significant growth) values or the integer 0 otherwise.

**Dataset 2**. We accessed the BacDive[56] API by using the R package *BacDive* (Vers. 0.8.0) to download for all available species data values of the subcategories "Enzymes", "Metabolite Utilization", and "Metabolite Production" in the category "Physiology and metabolism". This resulted in a matrix of 5289 attributes for 22,535 strains (as of 30th of November 2022). We noticed that the matrix is very sparse (99.3% missing values) and decided to focus on a subset of complete measurements concerned with fermentation phenotypes on different carbon sources (identified by "builds acid from …"). The final matrix used for our MASS analysis encompasses the fermentation of 637 species on 46 different carbon sources. Scripts used to mine the database and data itself are available on the project's GitHub repository (https://github.com/segrelab/MASS).

**Dataset 3**. Values from the table in Chapter 6 of ref. 27 were digitized into a table with 590 yeast strains (samples, $n$) and 92 phenotypes (Supplementary Data 2). We focused in this study on the 44 phenotypes representing growth on specific carbon sources (attributes, $m$). Symbols for positive ("+"), negative ("–"), delayed ("D"), weak ("W"), variable ("V"), and unknown ("?") growth were cast into categorical numbers. Negative growth was cast to 0; variable, weak, or delayed growth was cast to 1; positive growth was cast to 2; and unknown values were cast as null ("NaN"). Spearman correlations between each pair of features were calculated, and one of two highly correlated features (absolute Spearman correlation greater than 0.74) were dropped (Supplementary Data 3), reducing the number of features to 38. Lastly, samples with at least one missing (null) value reduced the number of samples to 462.

For MASS analysis, growth on D-Glucose was excluded as almost all species exhibited a positive phenotype. Furthermore, the sample "*Saccharomycodes sinensis*" was dropped as this species did not exhibit growth on any of the carbon sources. Hence, the phenotype matrix used for MASS consisted of 461 samples and 37 attributes. Categorical attributes were encoded as a one-hot numerical array, where each categorical level (negative; variable, weak, or delayed; and positive) was separately encoded and indicated by a 1 when that type of growth was exhibited. However, we eliminated the negative (0) categorical level, thereby yielding two dummy variables out of the three categorical levels and doubling the number of features to 74. Zero values were cast to $-1$ so that the categorical one-hot arrays had values of $-1$ and 1, which is appropriate for hinge loss.

## Linear programming approach to linear regression

Linear regression is a statistical method to model the linear relationship between the $i$'th response variable of $n$ samples, $y_i$ (a scalar), and the predictor variables of $m$ attributes, $\boldsymbol{x_i}$, (an $m \times 1$ vector), by estimating the parameters, $\boldsymbol{\beta}$, (an $m \times 1$ vector), that provides the best explanation for the data:

$$y_i = \boldsymbol{x_i'}\boldsymbol{\beta} + \varepsilon_i, \tag{1}$$

where the prime denotes transpose. One approach to model fitting involves minimizing the difference between the actual and predicted responses (the error). If the $\ell_1$-norm, $\sum_{i=1}^{n}|\hat{y}_i - y_i|$ is used as an error metric, then linear regression can be formulated as an optimization problem:

$$minimize_\beta \sum_{i=1}^{n}|y_i - \boldsymbol{x_i'}\boldsymbol{\beta}| \tag{2}$$
$$such \ that -M \le \beta_j \le M, \forall j,$$

where $M$ is a scalar value that bounds the estimated coefficients.

## A method to determine predictors and responses

We use the convention that all vectors are column vectors. We first define the $n \times m$ matrix, $\boldsymbol{X} = (X_{i,j}) = (\boldsymbol{x_1'}, \ldots, \boldsymbol{x_n'})'$, of observations with $n$ samples and $m$ attributes. We want to select a subset $p$ from the attributes and use this subset as predictors for the remaining $m - p$ attributes (responses):

$$\boldsymbol{X} = \boldsymbol{XB} + \boldsymbol{B_0} + \boldsymbol{E}, \iff \boldsymbol{x_i'} = \boldsymbol{x_i'}\boldsymbol{B} + \boldsymbol{\beta_0'} + \boldsymbol{\varepsilon_i'} \tag{3}$$

where $\boldsymbol{B} = (B_{i,j}) = (\boldsymbol{\beta_1}, \ldots, \boldsymbol{\beta_m})$ is the $m \times m$ coefficient matrix, $\boldsymbol{B_0} = (\boldsymbol{\beta_0'}, \ldots, \boldsymbol{\beta_0'})'$ is the $n \times m$ constant matrix, $\boldsymbol{\beta_0} = (\beta_{0,1}, \ldots, \beta_{0,m})'$ is the constant vector and $\boldsymbol{E} = (\boldsymbol{\varepsilon_1'}, \ldots, \boldsymbol{\varepsilon_n'})'$ is the noise matrix. Note that $(\boldsymbol{\beta_j})_k = B_{k,j}$ represents how attribute $j$ is used to predict attribute $k$, and $x_{i,j}$ denotes the value of attribute $j$ of sample $i$. Using the $\ell_1$-penalty as a loss function and including a sparsity constraint, Equation (3) can be formulated as a mixed integer linear program:

$$minimize_{B,\beta_0,w,z,t} \frac{1}{n}\sum_{i=1}^{n}\sum_{j=1}^{m} w_{i,j} + \lambda\sum_{j=1}^{m}\|\boldsymbol{\beta_j}\|^1$$

$$such \ that \quad t_{i,j} - \boldsymbol{x_i'}\boldsymbol{\beta_j} - \beta_{0,j} \le w_{i,j}, \forall i,j,$$

$$-t_{i,j} + \boldsymbol{x_i'}\boldsymbol{\beta_j} + \beta_{0,j} \le w_{i,j}, \forall i,j,$$

$$\sum_{j=1}^{m} z_j \le p,$$

$$-Mz_k \le (\boldsymbol{\beta_j})_k = B_{k,j} \le Mz_k, \forall j,k,$$

$$-Mz_j \le t_{i,j} - X_{i,j} \le Mz_j, \forall i,j,$$

$$w_{i,j} \ge 0, \forall i,$$

$$z_j \in \{0,1\}, \forall j,$$

$$B_{i,j} \in R, \beta_{0,j} \in R, \forall i,j, \tag{4}$$

where $\boldsymbol{w}$ is a dummy variable for the loss, reformulating the absolute loss using linear constraints; $\lambda$ controls the sparsity or robustness constraint; $\boldsymbol{t}$ is a dummy variable for whether an attribute affects the loss; $z$ is the indicator variable for whether an attribute is a predictor ($z_j = 1$) or a response ($z_j = 0$); $\boldsymbol{x_i}$ is the vector of the attributes of sample $i$; and $\boldsymbol{\beta_j}$ is the vector of coefficients to predict attribute $j$. If an attribute is a predictor, then coefficients are bounded between $\pm M$ and $t_{i,j}$ will take a value so that $w_{i,j}$ is set to zero, thus not affecting the loss. However, if an attribute is a response, then coefficients are set to 0 and $t_{i,j}$ is set to $X_{i,j}$, therefore affecting the loss.

Since our datasets involved categorical data, **Problem (4)** was further reformulated to use hinge loss for $X_{i,j} \in \{-1,1\}$[66]:

$$minimize_{B,\beta_0,w,z} \frac{1}{n}\sum_{i=1}^{n}\sum_{j=1}^{m} w_{i,j} + \lambda\sum_{j=1}^{m}\|\boldsymbol{\beta_j}\|^1$$

$$such \ that \quad \sum_{j=1}^{m} z_j \le p,$$

$$-Mz_k \le (\boldsymbol{\beta_j})_k = B_{k,j} \le Mz_k, \forall j,k,$$

$$X_{i,j}(\boldsymbol{x_i'}\boldsymbol{\beta_j} + \beta_{0,j}) \ge 1 - w_{i,j} - Mz_j, \forall i,j,$$

$$w_{i,j} \ge 0, \forall i,$$

$$z_j \in \{0,1\}, \forall j,$$

$$z_j = z_k \ if \ (j,k) \ are \ in \ one \ group,$$

$$B_{i,j} \in R, \beta_{0,j} \in R, \forall i,j. \tag{5}$$

MASS is implemented in *Python* using the MILP solver *GUROBI*.

## Heuristics solutions to speed up solution time

The MILP **Problem (5)** is NP-complete. We observed that solving **Problem (5)** with a larger $p$ (relatively close to the number of total attributes $m$) can be solved relatively quickly (on the order of minutes). However, solving this problem for fewer predictors becomes very slow or impossible. Consequently, we used a similarity-based heuristic method first to obtain a near-optimal feasible solution and offer it to the solver. This reduced the time needed by the solver to reach an optimal solution.

We observed a similarity between the predictors selected at constraint bounds $p$ and $p + 1$, where $p + 1$ is a larger constraint bound. Consequently, after solving **Problem (5)** at the constraint bound $p + 1$, the selector indicator variable is $z_{p+1}$. When solving at the constraint bound $p$, we added an additional constraint to ensure that the predictors at the constraint bound $p$ are selected from the predictors at the constraint bound $p + 1$:

$$z'_{p+1} z \geq p, \tag{6}$$

where $z$ is the selector indicator at the constraint bound $p$. After solving **Problem (5)** with the additional constraint, we obtain a sub-optimal selection indicator vector $z_{greed1}$. This first heuristic solution performs a monotone selection since we remove a single attribute from $z_{p+1}$.

In an alternative heuristic, we replaced **Constraint (6)** with the following relaxed version:

$$z'_{p+1} z \geq p - 1, \tag{7}$$

using $z_{greed1}$ as the starting point of the resulting MILP.

After solving **Problem (5)** with **Constraint (7)**, we obtain a sub-optimal selection indicator vector $z_{greed2}$. In all experiments discussed in this paper we used $z_{greed2}$.

## Statistics and reproducibility: assessing the attribute selection by MASS using random forest models

We used random forest (RF) classifiers as implemented in the Python package *scikit-learn* to assess the performance of attribute sets selected through MASS. Prior to training random forest models, we randomly selected 50% of the samples in our dataset to form the training and validation set and retained the remaining 50% of the samples as the test set. Threefold (for Dataset 3) and fivefold (for Dataset 1 and 2) cross-validation was used to tune parameters (e.g., the number of variables randomly sampled as candidates at each split, the maximum depth of the tree, etc.). The number of RF trained is determined by the number of allowed predictor variables because this also determines the number of available response variables. For example, given that a dataset consists of 11 variables and 1 predictor is allowed ($p = 1$), MASS will identify $(11-1) = 10$ response variables. For each of these response variables, one RF model is trained using the 1 predictor, which is identified by MASS. Similarly, if ten predictors are allowed ($p = 10$), MASS will identify $(11-10) = 1$ response variable for which one RF model will be trained using all the 10 predictor variables selected by MASS. To assess the performance of the predictor selections by MASS we build additional RF models with predictor sets either selected randomly to generate baseline performance values or using a predictor set selected by maximal entropy. The choice of the latter was motivated by the observation that MASS predictor selection preference correlated with Shannon entropy value for the respective attributes and served as a comparison of MASS to a more naïve predictor set selection approach.

We compared the following performance metrics: Accuracy, Balanced Accuracy, Cohen's kappa coefficient, Weighted macro-averaged F1 score, Jaccard distance, and Matthews correlation coefficient (MCC).

These metrics were applied as implemented in the Python package *scikit-learn* and were calculated after combining results from all RF models for one specific $p$ and selection method.

For our main results, we decided to use the Matthews Correlation Coefficient because this metric has been shown to give robust estimates even for unbalanced datasets[55]. However, for comparison, we also show the remaining set of performance metrics in the Supplementary Information (Supplementary Figs. 2, 3, 4).

## Reporting summary

Further information on research design is available in the Nature Portfolio Reporting Summary linked to this article.

## Data availability

Data generated and analyzed in this study are available in the Supplementary Information and the GitHub repository (https://github.com/segrelab/MASS; https://doi.org/10.5281/zenodo.10723101) accompanying this study.

## Code availability

The MASS algorithm and corresponding codes used for analyses are open source and available in the GitHub repository (https://github.com/segrelab/MASS; https://doi.org/10.5281/zenodo.10723101).

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

## Acknowledgements

IChP acknowledges funding by the NSF under grants CCF-2200052, DMS-1664644, and IIS-1914792, by the ONR under grant N00014-19-1-2571, by the DOE under grant DE-AC02-05CH11231, by the NIH under grant UL54 TR004130, and by Boston University. DS acknowledges funding by the BU Kilachand Multicellular Design Program, the U.S. Department of Energy, Office of Science, Office of Biological & Environmental Research through the Microbial Community Analysis and Functional Evaluation in Soils Science Focus Area Program (m-CAFEs) under contract number DE-AC02-05CH11231 to Lawrence Berkeley National Laboratory, the NIH National Institute on Aging award number UH2AG064704, the NSF Center for Chemical Currencies of a Microbial Planet (C-CoMP publication #039), NSF-BSF grant 2246707, and the Human Frontiers Science Program (grant numbers RGP0020/2016 and RGP0060/2021).

## Author contributions

D.S. and I.C.P. designed and supervised the study. M.T., T.W., D.S., and I.C.P. conceived the initial concept for the algorithm. T.W. and I.C.P. implemented the MASS algorithm. K.H., T.W., E.J.F., and M.T. applied the MASS algorithm to the various datasets and analyzed the data. T.W., M.T., I.C.P., E.J.F., and D.S. wrote the initial version of the manuscript. K.H. implemented the final statistical analysis, performed revisions with help from T.W., and wrote the final version of the manuscript with D.S. All authors discussed the results, revised the draft manuscript, and read and approved the final manuscript.

## Competing interests

The authors declare no competing interests.
