## [Peer Review File · Communications Biology]

Reviewers' comments:

Reviewer #1 (Remarks to the Author):

Dear Editors,

the submitted manuscript proposes an approach to address the optimal phenotype data management in a biological context for the task of finding an optimal subset of features for different biological conditions. I recognize the importance of the topic above all in the latest years when a huge amount of such phenotypes information are extracted from living organisms' understanding and control.

Although the paper is well written and organized, and the problem well motivated, I found some critical issues that have to be solved before the manuscript is revised again, going into major revision.

Major concerns.

The authors repeatedly underline the fact that dataset 1 they use as a proof of concept of a binary classification problem, is small in size, possibly not fully representative, and skewed in the distribution of values. this assertion makes me in doubt of the general validity of the approach above all when conclusions are in conflict with those extracted by dataset 2 results. I think that it is fundamental to include a third dataset to the analysis as large as the dataset2 that encompasses such criticism and that confirm the results obtained using dataset 2 about the general validity of the MOSS/RF architecture.

About the novelty introduced in the work, It is not clear which are the advancements with respect to the state of the art. In particular, with respect to MOSS strategy, there are similar approaches with a similar name in recent literature, named "MOSS—Multi-Modal Best Subset Modeling in Smart Manufacturing." Despite the different fields of application, it is important to stress the difference in determining a subset of descriptors in the data classification task.

The authors should list the advancements they propose with respect to the state of the art in a clear and referenced way.

Minor concerns.

Regarding microfluidic devices already used in a similar context and the investigation with phenomics, please expand the references (Cario, 2022). consider for example:

"PhenoChip: A single-cell phenomic platform for high-throughput photophysiological analyses of microalgae." *Science advances* 6.36 (2020): eabb2754."

"Machine learning phenomics (MLP) combining deep learning with time-lapse-microscopy for monitoring colorectal adenocarcinoma cells gene expression and drug-response." *Scientific Reports* 12.1 (2022): 1-14."

"Phenomics approaches to understand genetic networks and gene function in yeast." *Biochemical Society Transactions* 50.2 (2022): 713-721."

line 126. insert a ref and provide a sketch of the index

line 241. See the major concerns above mentioned.

line 443. This is the so-called unbalanced accuracy since it does not take into account the possible unbalance of class data. Please add also a balanced accuracy index.

References. Improve as suggested.

Reviewer #2 (Remarks to the Author):

The paper by Forchielli et al. proposes a computational method to predict phenotypes produced by a set of environmental conditions, followed by a machine learning technique (Random Forest). The method tries to answer the following question: if we can choose only a subset of the environmental conditions, and aim at using phenotypes under such conditions as predictors of phenotypes under all other conditions, what is the ideal choice of conditions we should choose? This method is then applied to two microbial datasets to predict the population-level fitness (growth rate) of phenotypes in a series of experiments that manipulated resource types in cultures.

Major comments

1) The paper is quite technical in its language and when describing the results. Given that Communications Biology is a general biology journal, I highly suggest you to reformulate the way you present the results and also the description of the method to make it more understandable for biologists.

2) I'd appreciate if you compare MOSS with other related methods in the Discussion.

Minor comments:

1) Delete L. 203-5

Reviewer #3 (Remarks to the Author):

The article describes the use of the Multi-output subset selection algorithm to select useful characteristics. This study allows to reduce the number of experiments needed to identify a strain or to map the metabolic pathways. It also develops a field of microbe phenomics, which is modern and important area of systems biology. Paper is well-written and has some sound data. Below are the points that the authors should address for the revision.

1. «The number of RF models trained depend on the number of predictors to which MOSS has been constrained: for example, when applied to a dataset containing 11 phenotypes, if p was set to 1, 10 random forest models were trained, and when p was set to 10 predictors, only 1 random forest model was trained.»

Query: Please describe in detail (and ideally incorporate to the ms) why this exact approach was used for the selection.

2. «The first dataset (DATASET 1) is a matrix of growth phenotypes (optical density (OD)) for 65 marine bacteria on 11 different media» –

Query: It makes sense to describe the databases (the number of instances and the balance of classes, and so on).

3. «Accuracy Score=((True Positives+True Negatives))/((True Positives+True Negatives+False Positives+False Negatives).)» –

Query: What is the reason for the choice of this particular metric and is there any information about comparing this metric with others (F-score, AUC, etc.)?

We are grateful to the Reviewers for the constructive comments on our manuscript. In order to properly address all comments, we did several major revisions to the manuscript and the work itself. Most importantly, we added a new dataset to corroborate our results, and we drastically revised the strategy to assess the performance of MASS-selected attributes to provide a statistically sound validation of the random forest models. Our changes to the manuscript are reported in the detailed point-by-point response below.

Reviewer #1:

Dear Editors,

the submitted manuscript proposes an approach to address the optimal phenotype data management in a biological context for the task of finding an optimal subset of features for different biological conditions. I recognize the importance of the topic above all in the latest years when a huge amount of such phenotypes information are extracted from living organisms' understanding and control.

Although the paper is well written and organized, and the problem well motivated, I found some critical issues that have to be solved before the manuscript is revised again, going into major revision.

Major concerns.

R1.1 The authors repeatedly underline the fact that dataset 1 they use as a proof of concept of a binary classification problem, is small in size, possibly not fully representative, and skewed in the distribution of values. this assertion makes me in doubt of the general validity of the approach above all when conclusions are in conflict with those extracted by dataset 2 results. I think that it is fundamental to include a third dataset to the analysis as large as the dataset2 that encompasses such criticism and that confirm the results obtained using dataset 2 about the general validity of the MOSS/RF architecture.

We are grateful to the reviewer for pointing out these shortcomings and for the excellent suggestion to include an additional dataset. We have significantly revised the manuscript to address this criticism as reflected in revised figures, Results (Page 5, Lines 180-194) and Discussion (Page 10, Lines 401-418) sections.

In particular, we have identified a third major dataset (now labeled DATASET 2) that we felt was similar in nature to DATASET 1 (discrete, binary variables), but of comparable size (637 species under 46 conditions) to what is now labeled DATASET 3 (formerly DATASET 2, 492 species under 38 conditions). The abstract was updated accordingly (Page 1, Line 29). The new DATASET 2, obtained from one of the largest repositories of bacterial phenotypic data (BacDive, <https://bacdive.dsmz.de/>) pertains to fermentation phenotypes of bacteria on different carbon sources. The new analysis is reported in the revised manuscript under section "Analysis of DATASET 2: Bacterial fermentation on different carbon sources" (Pages 6-7, Lines 236-293),

and illustrated in the new Figure 3. Importantly, our key results are now supported by more appropriate metrics, and they are consistent across the large datasets (DATASET 2 and 3). In particular: (1) We now assess the success of a random forest that uses MASS-predicted features through a new performance metric that takes into account possible skewness in the distribution of data (see Methods section “Assessing the attribute selection by MASS using random forest models” at Pages 14-15, starting Line 547) [See also responses to R1.6 and R3.3]; (2) In our revised figures, we systematically compare the performance of our method to an alternative entropy-based heuristic, as well as to randomly selected features, demonstrating the broad applicability of MASS; (3) We now discuss the observed general trend of the MASS-based random forest performance which increases with the number of MASS-selected predictors used for training. At some point, as the environments with high informational value are exhausted, the performance starts decreasing. The interpretation of this trend is discussed in detail at Page 10, Lines 401-418. (4) A newly added figure (Figure 3E-G on Page 24) highlights the fact that the top environments that MASS chooses as predictors correspond to the main entry point of carbon in central metabolism, demonstrating that MASS can also help reconstruct the hierarchy of key biological processes associated with a phenotypic dataset. (5) We clarify that, while here we use MASS in conjunction with a random forest, features selected using MASS can be in principle used with any supervised learning method. We adapted the first subsection of the Result section (Page 5, Lines 180-183) and Figure 1 to reflect this notion.

Finally, while, as pointed out by the Reviewer, DATASET 1 is not an ideal benchmark for the general validity of MASS due to its small sample size, we still feel it is a useful proof-of-principle to illustrate the concept, and it serves as a preliminary validation because it recapitulates a previously reported biological interpretation.

R1.2 About the novelty introduced in the work, It is not clear which are the advancements with respect to the state of the art. With respect to MOSS strategy, there are similar approaches with a similar name in recent literature, named "MOSS—Multi-Modal Best Subset Modeling in Smart Manufacturing." Despite the different fields of application, it is important to stress the difference in determining a subset of descriptors in the data classification task.

The authors should list the advancements they propose with respect to the state of the art in a clear and referenced way.

We thank the reviewer for this helpful comment which prompted us to distill and clarify for the readers what makes MASS unique relative to existing approaches. Our work is motivated by the general question of how to select the subset of environmental conditions in each phenotypic dataset, which best describe the dataset.

The most important message, now described in a thoroughly revised Introduction (Pages 3-4, Lines 104-134), Results (Page 5, Lines 139-154) and Discussion (Page 9, Lines 352-358), is that most existing methods are either focused on identifying the most salient features of a dataset (e.g., PCA), or dedicated to finding the best relationship (e.g., through linear regression) between a set of variables labeled as input variables and a set of variables labeled as output variables. In MASS these two aspects are combined into a single novel algorithm: **given a**

matrix of phenotypes under n conditions, with no a-priori distinction between inputs and outputs, the algorithm selects the p conditions that are best at predicting through a linear relationship the remaining $n-p$ conditions. What is unique is that our MILP formulation simultaneously scans all possible combinations of p variables (for all values of p between 1 and n) and performs linear regression based on such variables over multiple outputs. MASS is also motivated, and its advancements are mentioned in the abstract (Page 1, Lines 20-27).

Many apparently similar approaches (including the MOSS from Sensors 2021, 21(1), 243) explore different flavors of generalized linear regression, and can search for the most important input variables, but based on an already existing distinction between input and output variables. In addition to capturing the essence of the novelty of our approach, we have provided additional background on prior approaches with corresponding references, including to the MOSS article pointed out by the Reviewer.

To highlight for the readers right away the core novelty of our algorithm, we also substantially modified the Abstract to include a concise but much more informative and accurate description of the approach around which our article is centered.

Minor concerns.

R1.3 Regarding microfluidic devices already used in a similar context and the investigation with phenomics, please expand the references (Cario, 2022). consider for example:

"PhenoChip: A single-cell phenomic platform for high-throughput photophysiological analyses of microalgae." Science advances 6.36 (2020): eabb2754."

"Machine learning phenomics (MLP) combining deep learning with time-lapse-microscopy for monitoring colorectal adenocarcinoma cells gene expression and drug-response." Scientific Reports 12.1 (2022): 1-14."

"Phenomics approaches to understand genetic networks and gene function in yeast." Biochemical Society Transactions 50.2 (2022): 713-721."

We thank the reviewer for the specific suggestions for additional references. We have added all the references recommended by the reviewer, as well as a citation to another related important study concerned with a microfluidic device for microbial phenotyping (Kehe'2021, Science Advances).

R1.4 line 126. insert a ref and provide a sketch of the index

Given that the line the reviewer points to refers to the hinge loss function, we have now added clarifying text about this aspect of our algorithm and added a reference as requested. In the Results section, where this mention used to be, we now explain in a less technical way how we evaluate our approach with random forest models trained independently using the MASS-selected attribute sets. This is now done through an improved performance score that corrects for potentially imbalanced data and compared to attribute sets selected alternatively. We assume that this is what the reviewer refers to when mentioning the "index"; please let us know if something else was meant. In addition, in the Methods section, where the hinge loss function

is described (Page 13, Lines 514-515), we have now added a reference to an article titled “Learning sparse classifiers: continuous and mixed integer optimization perspectives”, by Dedieu et al.

R1.5 line 241. See the major concerns above mentioned.

This line pertains to the skewness of values, and possible confounding effects of the small number of values for DATASET 1. We very much agree with the reviewer that the earlier interpretation needed further support and are grateful for the motivation to investigate this in more detail. As mentioned above [Point R1.1], we included a third dataset (now DATASET 2) to probe the generality of the MASS approach. We feel that the addition of the new dataset allowed us to provide a much more robust interpretation of our results and are grateful to the reviewer for prompting us to perform this additional analysis.

R1.6 line 443. This is the so-called unbalanced accuracy since it does not take into account the possible unbalance of class data. Please add also a balanced accuracy index.

The reviewer raised an important issue for which we are grateful as it led us to rethink the metrics for our analysis, and drastically modify main figures and text as described below. Of note, Reviewer 3 raised a similar issue, and we would also like to point to our discussion there [R3.3].

Indeed, the imbalance of class data can significantly affect measures of accuracy. In the revised manuscript we adopted a different metric for quantifying the capacity of our approach to provide accurate predictions. Specifically, we used as our main metric the Matthews Correlation Coefficient (MCC) which is a robust performance metric able to be applied to even highly unbalanced datasets [Chicco’2020, BMC Genomics]. We provide several additional performance measures in the supplement in case the reader wants to compare different metrics to each other. However, we would like to note that the general observations we discuss hold for all metrics which are considered relatively robust against class imbalance, namely balanced accuracy, Cohen’s Kappa score, and MCC. Moreover, prompted by this comment, we realized that it would be important to also compare, using the same metric, our approach to alternative naive choices of attributes, including (i) the usage of attributes based on their informational content (i.e., prioritizing variables with maximal Shannon entropy), as well as (ii) a completely random choice of attributes. We feel that this new analysis provides a much deeper and more transparent picture of our approach, including the interesting natural limit in the number of variables beyond which our approach becomes less significant, for reasons explained in detail in the text (Pages 7, Lines 284-293; Page 8, Lines 329-333 and Page 10, Lines 401-418).

R1.7 References. Improve as suggested.

We revised the references according to the reviewer’s suggestions.

Reviewer #2:

The paper by Forchielli et al. proposes a computational method to predict phenotypes produced by a set of environmental conditions, followed by a machine learning technique (Random Forest). The method tries to answer the following question: if we can choose only a subset of the environmental conditions, and aim at using phenotypes under such conditions as predictors of phenotypes under all other conditions, what is the ideal choice of conditions we should choose? This method is then applied to two microbial datasets to predict the population-level fitness (growth rate) of phenotypes in a series of experiments that manipulated resource types in cultures.

Major comments

R2.1 The paper is quite technical in its language and when describing the results. Given that *Communications Biology* is a general biology journal, I highly suggest you to reformulate the way you present the results and also the description of the method to make it more understandable for biologists.

We greatly appreciate this important comment and agree with the reviewer that it is important to convey in a less technical way the importance of our work. In revising the manuscript, we have done some major changes in the text of the Introduction to outline the aim and rationale for the development of MASS (Page 3, Lines 103-113). Most importantly, prompted by this valuable comment by the reviewer, we added a new section at the beginning of the Results (titled “A method to separate growth phenotypes into predictors and responses”, Page 4), aimed at explaining in non-technical terms the core idea of our algorithm. Finally, we have revised some paragraphs of the Discussion (Page 9, Lines 369-384) to more clearly (and less technically) delineate the possible implications of our algorithm in biology and other fields.

We report here a key portion of this new Result section, for the Reviewer's benefit:

“We designed MASS with the goal to explore large phenotypic datasets. Specifically, we want to choose the most informative subset of the environmental conditions and aim at using phenotypes under such conditions as predictors of phenotypes under all the other remaining conditions. The challenge of this question is that it involves two steps that are usually performed in separate calculations. In a first step we would typically choose which phenotypes are predictors (independent variables), and which are the response (dependent variables). In a second step, we would perform a linear regression, i.e., find coefficients necessary to compute the response phenotypes as linear combinations of the predictor phenotypes. What makes our approach novel and mathematically challenging, is that we do not know or assume a priori which phenotypes will be predictors and which ones will be responses. The MASS algorithm makes it possible to pursue both steps concurrently. In other words, the algorithm explores the many possible choices of predictors and simultaneously identifies the ones such that regression done using those predictors gives the best estimate of the responses. This algorithm involves both integer variables, describing which phenotypes are chosen as predictors, and continuous

variables which capture the regression coefficients, and is thus implemented using a Mixed Integer Linear Programming (MILP) approach.” (Page 4, Lines 139-154)

R2.2 I'd appreciate if you compare MOSS with other related methods in the Discussion.

We very much value this comment (also addressed in R1.2) and rewrote multiple portions of the manuscript to clarify the value of our methods in the context of existing approaches. Specifically, in the thoroughly revised Discussion, the opening paragraph is dedicated to comparing our approach to existing methods (Pages 8-9, Lines 337-358), and including relevant references. In addition, this comment prompted us to add more material on the novelty of our approach not only in the Discussion, but also in the Introduction and in the Results (see also response to previous point R2.1).

Minor comments:

R2.3 Delete L. 203-5

We have removed the text pointed to by the reviewer ("One consequence of using all variables to predict one response is overfitting; using all of the available data may introduce too much noise to the model, and therefore negatively affect performance."). In fact, the whole section on testing performance has been thoroughly re-written in response to other reviewers' comments [R1.6, R3.3], in conjunction with our newly introduced usage of a more rigorous metric and comparison with alternative naive feature selection methods. The behavior at large p (i.e., using many variables as predictors) is thoroughly discussed in the Discussion (Page 10, Lines 401-418).

Reviewer #3:

The article describes the use of the Multi-output subset selection algorithm to select useful characteristics. This study allows to reduce the number of experiments needed to identify a strain or to map the metabolic pathways. It also develops a field of microbe phenomics, which is modern and important area of systems biology. Paper is well-written and has some sound data. Below are the points that the authors should address for the revision.

R3.1 “The number of RF models trained depend on the number of predictors to which MOSS has been constrained: for example, when applied to a dataset containing 11 phenotypes, if p was set to 1, 10 random forest models were trained, and when p was set to 10 predictors, only 1 random forest model was trained.»

Query: Please describe in detail (and ideally incorporate to the ms) why this exact approach was used for the selection.

We are grateful for the opportunity to clarify this part of our work. As per the Reviewer's question, indeed our explanation was confusing. The essence is that since our algorithm has

multiple output variables, one needs to construct as many random forest classifiers as the number of output variables. The input variables chosen by MASS are incorporated as input variables in each of these random forest models. We have modified the text at Page 5, Lines 168-177 to clarify this point.

R3.2 «The first dataset (DATASET 1) is a matrix of growth phenotypes (optical density (OD)) for 65 marine bacteria on 11 different media» –

Query: It makes sense to describe the databases (the number of instances and the balance of classes, and so on).

We very much appreciate this important suggestion and took multiple steps to improve the paper along these lines: (i) We added in the Methods a paragraph describing each dataset and its origin; (ii) We have expanded the biological interpretation part of all our datasets, including, most prominently a newly introduced dataset on bacterial fermentation phenotypes. For this dataset we have also added a new figure (Fig. 3E-G), which provides a detailed biological interpretation of the results; (iii) We have significantly revised the statistical analysis to avoid issues with class imbalance in the different datasets. Correspondingly, we describe the class imbalance both in the Results section (Page 7, Lines 284-293) and in the Methods, where we introduced different metrics for assessing the performance of the random forest models under subsection “Assessing the attribute selection by MASS using random forest models” (Pages 14-15, starting at Line 547).

R3.3 «Accuracy Score= $((\text{True Positives} + \text{True Negatives}) / ((\text{True Positives} + \text{True Negatives} + \text{False Positives} + \text{False Negatives})))$ » –

Query: What is the reason for the choice of this particular metric and is there any information about comparing this metric with others (F-score, AUC, etc.)?

As also described in response to R1.6, we have significantly revised our performance analysis based on the observation that especially DATASET 2 and DATASET 3 are partially unbalanced (as it is very typical for biological datasets). We explored a variety of different metrics which are designed to account for imbalance in the data. In the end, we decided to use for our main results the Matthews Correlation Coefficient (MCC) which has been described to be a robust measure even for highly unbalanced data [Chicco, 2020, BMC Genomics] (Page 5, Lines 177-179). Nevertheless, we felt inclined to present all possible performance metrics we inspected in the supplement to show that the general trends we discuss are corroborated by most metrics. Still, it has to be noted that the breakdown of performance for $p > 15$ only becomes apparent for metrics robust against unbalanced data (i.e., balanced accuracy, Cohen’s Kappa score, and MCC). We are therefore very grateful for the reviewers’ comment which motivated us to further investigate these specifics.

REVIEWERS' COMMENTS:

Reviewer #1 (Remarks to the Author):

Dear Authors,

I recognize you did a lot of job to improve the manuscript according to the Reviewers comments. I consider that the manuscript in its actual form is suitable for publication.
Thank you.

Reviewer #2 (Remarks to the Author):

I am very satisfied with the way authors dealt not only with my previous, but also the other reviewer's. I congratulate them and recommend acceptance as is.

Reviewer #3 (Remarks to the Author):

The authors corrected MS accordingly to comments and clarified some points. The random forest algorithm has been refined. So that the input variables selected by MASS are included as inputs. The data sets were described and the origin is added. The statistical analysis was revised to avoid problems with class imbalance in different data sets.